# Brief Communication: An Electrifying Atmospheric River: Understanding the Thunderstorm Event in Santa Barbara County during March 2019

Deanna Nash[1] and Leila M.V. Carvalho[1,2]

[1]Department of Geography, University of California, Santa Barbara, CA 93106 USA
[2]Earth Research Institute, University of California, Santa Barbara, CA 93106, USA

**Correspondence:** Deanna Nash (dlnash@ucsb.edu)

**Abstract.** On 5 March 2019 12 UTC, an Atmospheric River (AR) made landfall in Santa Barbara, CA and lasted approximately 30 hours. While ARs are typical winter storms in the area, the extraordinary amount of lightning strikes observed near coastal Santa Barbara made this event unique. The Earth Networks Global Lightning Network (ENGLN) detected 14,416 lightning flashes in southern California (20°N to 50°N and 140°W to 110°W) in 24 hours, which is roughly 1500 times the climatological flash rate in this region. The AR related thunderstorm resulted in approximately 23.18 mm accumulated precipitation in 30 hours in Santa Barbara. This article examines synoptic and mesoscale features conducive to this electrifying AR event, characterizing its uniqueness in the context of previous March events that made landfall in the region. We show that this AR was characterized by an unusual deep moist layer extending from low-to-mid troposphere in an environment with potential instability and low elevation freezing level. Despite the negligible convective available potential energy (CAPE) during the peak of the thunderstorm near Santa Barbara, the lifting of layers with high water vapor content in the AR via warm conveyor belt and orographic forcing in a convectively unstable atmosphere resulted in the formation of hail and enhanced electrification.

## 1 Introduction

Due to recent wildfire activity in Santa Barbara County (e.g. Thomas Fire during December 2017, Whittier Fire during July 2017, and Sherpa Fire during June and July 2016) this region is at high risk for post-fire debris flow when 15 minutes of rainfall has an intensity greater than or equal to 24 mm hour[-1] (USGS, 2019). These conditions were observed during the devastating Montecito debris flow on the 9 January 2018 that resulted in 23 deaths, 246 structures destroyed, and 167 damaged structures (Oakley et al., 2018). On 1 March 2019, the National Weather Service (NWS) in Oxnard, CA forecasted 2 storms to hit Santa Barbara County (1-2 March 2019 and 5-6 March 2019). On 5 March 21 UTC, a mandatory evacuation order was issued for the Thomas, Whittier, and Sherpa fire burn areas due to the prediction of a subsequent severe storm and flood potential that existed for low-lying areas given increased ground saturation from the storm on 2 March 2019, impacting about 3,000 residents. While

no significant debris flows were triggered during this event, a combination of an Atmospheric River (AR) and an extreme number of lightning strikes made this storm exceptional. Figure 1a shows a photo of lightning strikes at the Santa Barbara Harbor during the storm taken by Santa Barbara County Fire Department's Mike Eliason.

The term Atmospheric River (AR), describes a phenomenon that explains how baroclinic eddies transport large amounts of water vapor via relatively infrequent, long conduits of strong moisture transport across mid-latitudes and into polar regions (Newell et al., 1992; Zhu and Newell, 1994). Many studies have focused on the regional impacts of ARs in western United States and have found that ARs bring large amounts of moisture to the west coast of North America and are related to precipitation extremes and flooding, particularly in the winter season (Harris and Carvalho, 2018; Dettinger, 2011; Guan et al., 30    2010, 2013; Ralph et al., 2006). Despite occurring less frequently than ARs in Northern California, the Southern California ARs have significant impact in the hydrological cycle of the region (Harris and Carvalho, 2018; Cannon et al., 2018; Oakley et al., 2018; Oakley and Redmond, 2014). Although ARs are often associated with extreme precipitation, flooding, and other hazardous events, they play critical role in replenishing reservoirs and underground water resources, particularly in dry areas of Southern California. Studies show that just a few AR events each year can contribute the majority of the precipitation and 35    streamflow that regulates the state's water resources (Cannon et al., 2018; Gershunov et al., 2017; Ralph et al., 2019; Dettinger, 2013). Ralph et al. (2019) have developed a scale to characterize ARs based on intensity and duration, pointing out that ARs can result in a wide spectrum of conditions from beneficial to hazardous. As of now, no studies have examined the relationship between ARs and lightning.

Lightning usually occurs when the electric charges in a cloud separate and exceed the intensity that the air can sustain (Price, 40    2013). Charges usually build up in the mixed phase region of the clouds ($0°C$ to $-40°C$) when there are enough updrafts to lift particles above the freezing level (Price and Rind, 1993). The correlation between cloud-top height and lightning rate is well documented and can be attributed to the deep vertical development of convective thunderstorms (Price and Rind, 1993; Pessi and Businger, 2009). Pessi and Businger (2009) documented that lightning activity can be associated with cold temperatures aloft or convection along cold fronts.

Although the thunderstorms on 5 March 2019 caused minimal damage (e.g. small lightning fires, power outages), this event was meteorologically significant because of the exceptional number of lightning strikes in such a short period. This study examines synoptic and mesoscale dynamics, as well as the thermodynamic characteristics of this AR and investigates the uniqueness of this event compared to past March ARs that made landfall in Santa Barbara.

## 2   Data and Methods

Climate Forecast System version 2 (CFSv2) (Saha et al., 2014) operational analysis was used in this study to evaluate the synoptic and mesoscale meteorological conditions between $10°N$ and $50°N$ and $150°W$ to $110°W$ between the dates 4 March 2019 18 UTC and 6 March 2019 18 UTC. CFSv2 data at $0.5° \times 0.5°$ horizontal resolution was obtained at 37 pressure levels between 1000 hPa and 1 hPa at a 6-hourly time scale. AR conditions are determined based on IVT (see appendix for calculation) exceeding 250 kg m$^{-1}$ s$^{-1}$ at a fixed geographical point. The AR event in this study refers to the time that the AR conditions

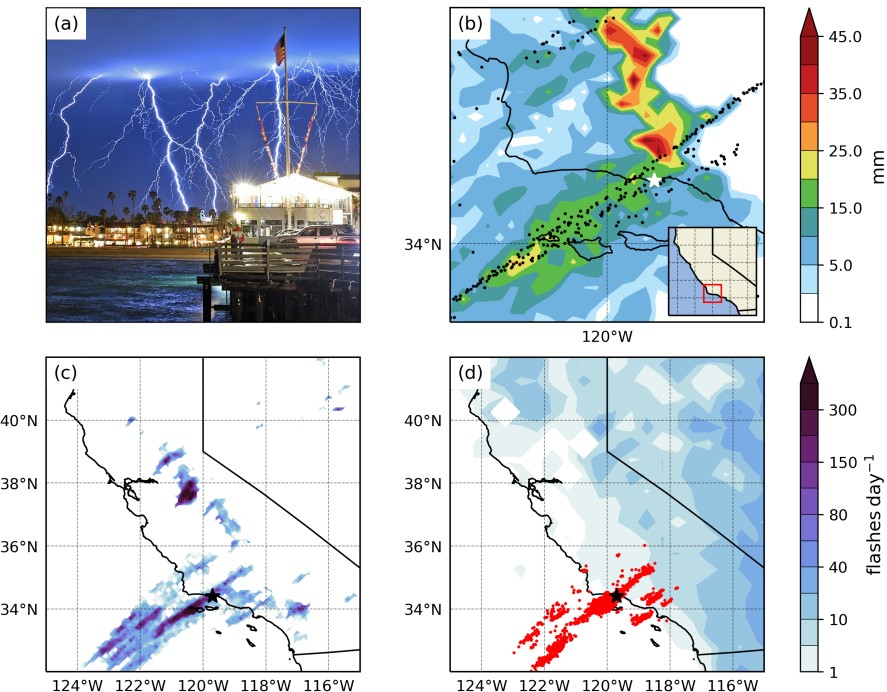

**Figure 1.** (a) Photo of lightning at the Santa Barbara Harbor in Santa Barbara, CA taken by Mike Eliason from the Santa Barbara County Fire Department during the storm at 6 March 2019 4 UTC. (b) NOAA NEXRAD L3 precipitation accumulation (shaded; mm) and locations of NOAA NEXRAD L3 Hail Signatures (black points) between 5 March 2019 12 UTC and 6 March 2019 23:59 UTC. The location of Santa Barbara is indicated by the white star. (c) ENGLN lightning strike frequency (shaded; flashes day$^{-1}$) on 6 March 2019. The location of Santa Barbara is indicated by the black star. (d) Climatological annual mean lightning density (shaded; flashes day$^{-1}$) between 1995 and 2014 using TRMM LIS-OTD lightning climatology and lightning strike locations (red points) between 4 and 5 UTC on 6 March 2019 based on ENGLN. The location of Santa Barbara is indicated by the black star.

occurred in Santa Barbara, i.e. at the grid cell centered on 34.5°N and 119.5°W. The duration of the AR event is determined by the time (in hours) that the AR conditions are consecutively met. NASA's Modern Era Retrospective Reanalysis Version 2 (MERRA2) (Gelaro et al., 2017b; Bosilovich et al., 2015) and the global atmospheric river detection catalog that identifies atmospheric rivers on a global, 6-hourly basis were used to determine the anomalous characteristics of the March 2019 AR event compared to past ARs that made landfall in Santa Barbara. This AR detection algorithm was introduced in Guan and Waliser (2015) and refined in Guan et al. (2019). Here we analyzed ARs and their characteristics on a daily temporal scale at 0.5° by 0.625° spatial resolution between 1980 and 2018. The other calculated variables from CFSv2 are dew point (Td) and equivalent potential temperature ($\theta_E$), which are calculated based on Bolton (1980, eq. 11, 43).

Lightning flash data obtained from Earth Networks Global Lightning Network (ENGLN) (Earth Networks, 2019) was used to quantify the location and number of lightning strikes between 4 March 2019 0 UTC and 7 March 2019 0 UTC near southern California. The global lightning network, which includes more than 1,700 sensors, detects lightning flashes and provides

various information about those flashes, including latitude, longitude, amplitude of the lighting, duration of the flash, and the number of in-cloud (IC) and cloud-to-ground (CG) lightning pulses within a given flash (Earth Networks, 2019). A lightning flash can be made up of one or more IC or CG lightning pulses, which connect regions of opposite polarity. To put the extremity of this lightning event into climatological context, an annual lightning strike climatology from Tropical Rainfall

Measuring Mission Lightning Imaging Sensor and Optical Transient Detector (TRMM LIS-OTD) (Cecil, 2015) was used at a horizontal resolution of 0.5° by 0.5° between 1995 and 2014 for the region surrounding southern California (20°N to 50°N and 140°W to 110°W). Comparing the two lightning sources has a certain level of uncertainty, since TRMM LIS-OTD and ENGLN do not overlap temporally. However, because this event had significantly above average lightning flash rates compared to the climatology, the possible error introduced by comparing two different data sets does not impact the results.

For precipitation, hourly precipitation data from National Oceanic and Atmospheric Administration's Next Generation Radar Level 3 (NOAA's NEXRAD L3) was used between the dates 4 March 2019 18 UTC and 6 March 2019 23:59 UTC at roughly 1 km resolution (NOAA National Weather Service (NWS) Radar Operations Center, 2019). To identify the approximate location, time, and diameter of hail, NOAA's NEXRAD L3 hail signature product was used. To identify cloud convection and cloud top height via cloud top temperature, the Cloud and Moisture Imagery (CMI) product from GOES-R (GOES-17) Advanced

Baseline Imager Level 2 was obtained for 6 March between 3 UTC and 5 UTC at 5-minute temporal intervals and 10 km by 10 km spatial resolution (GOES-R Algorithm Working Group and GOES-R Series Program, 2017).

## 3   Results and Discussion

### 3.1   March 2019 Event

An AR made landfall near Santa Barbara (34.5°N, 119.5°W) between 5 March 12 UTC and 6 March 18 UTC, resulting in

total accumulated precipitation of approximately 23 mm around Santa Barbara according to NOAA NEXRAD L3 one-hour precipitation. While this was not enough precipitation to initiate debris flow, instances of hail were identified by the NOAA NEXRAD L3 hail signature product (see Fig. 1b). The presence of hail indicates strong updrafts and a low freezing level, which are conditions that also favor the development of lightning in a storm (Pruppacher and Klett, 1997). During this AR event, ENGLN detected 73,442 flashes of lightning with 119,363 combined IC and CG pulses around Southern California (140°W

to 110°W and 20°N to 50°N) (Earth Networks, 2019). Among these, 14,416 flashes of lightning with 50,399 combined IC and CG pulses occurred in the 24 hour period following 6 March 0 UTC (Fig. 1c). TRMM LIS-OTD records an area annual average of 9.15 flashes per day in the region surrounding southern California (20°N to 50°N and 140°W to 110°W), making the 14,416 lightning flashes in under 24 hours very extreme. In fact, even if this was the only lightning activity for 2019, it would represent about 1,500 times the climatological rate (Fig. 1d) (Cecil, 2015). Based on the AR database of Guan et al.

(2019), on average 10 AR days are observed between December and March each year in Santa Barbara, with a total of 742 AR days associated with ARs that made landfall in the grid cells closest to Santa Barbara between 1980 and 2019. When compared to the TRMM LIS-OTD low resolution time series, between 1995 and 2014 there were approximately 350 landfalling AR events that coincided with lightning flashes, with the majority of events resulting in less than 60 flashes per day (Cecil, 2015).

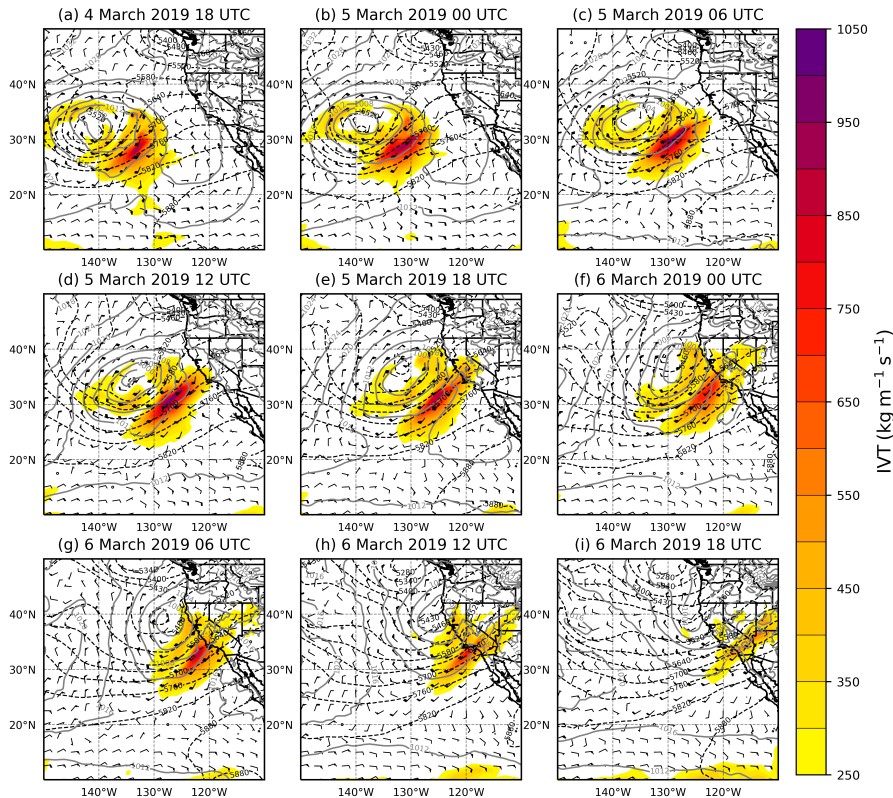

**Figure 2.** CFSv2 data showing IVT (shaded; kg m$^{-1}$ s$^{-1}$), 850 hPa wind (barbs; knots), Mean sea level pressure (grey contours; hPa), and 500 hPa geopotential height (black dashed contours; m) at 6-hourly time steps between 4 March 2019 18 UTC and 6 March 2019 18 UTC. The time step closest to the peak of the event is shown in figure (g) at 6 March 2019 6 UTC in the bottom left corner.

### 3.2 Extratropical Cyclone and AR Conditions

Following an extratropical cyclone that made landfall at 1 March 12 UTC, a deep mid-level (500 hPa) trough developed into a closed low system, forming a pool of cold air centered at approximately 32°N and 140°W by 4 March 18 UTC (Fig. 2a). The surface low-pressure was located directly below the 500 hPa closed low on 4 March 18 UTC (Fig. 2a). This mid-level closed low moved eastward and northward, until 6 March 12 UTC when it was no longer closed (Fig. 2h). According to Oakley and Redmond (2014), 41-50% of precipitation in Santa Barbara between October and March is associated with closed lows. The surface low-pressure deepened from 1005 hPa to approximately 996.36 hPa by the peak event time at 6 March 2019 6 UTC; at this point it was centered around 38°N and 126°W, west of northern California (Fig. 2g). At the peak time of the event, 6 March 6 UTC, the jet streak exit region was located at 35°N and 122°W, directly northwest of Santa Barbara (Fig. S1g). Pessi and Businger (2009) showed that most of the storms that have lightning activity over the North Pacific Ocean are associated with similar synoptic conditions as those observed during the storm in March 2019.

These synoptic conditions provided the dynamical mechanisms necessary for subtropical moisture to be transported via an AR, shown as the area of IVT greater than 250 kg m$^{-1}$ s$^{-1}$ (Fig. 2). This AR made landfall at approximately 5 March 12 UTC on the west coast near Santa Barbara and lasted approximately 30 hours (Fig. 2d-i). The peak IVT value for this event within the AR was 1034 kg m$^{-1}$ s$^{-1}$ at 5 March 12 UTC (Fig 2d). In the grid cell closest to Santa Barbara (34.5°N and 119.5°W) the AR had a peak IVT value of 446 kg m$^{-1}$ s$^{-1}$ on 6 March 6 UTC (Fig. S2a). Based on the duration (30 hours) and maximum

instantaneous IVT intensity of the AR (446 kg m$^{-1}$ s$^{-1}$), this event is categorized at AR-CAT 1 according to Ralph et al. (2019), indicating that this AR was most likely beneficial to the Santa Barbara area. This particular AR had IVT direction and magnitude characteristics similar to past ARs that made landfall in the Santa Barbara area (Fig. S2b, S2c).

     Equivalent potential temperature at 850 hPa ($\theta_E$) (Fig. S3) identifies the formation of the warm conveyor belt (WCB), or the ascending air within the warm sector of the extratropical cyclone and the overlap of the AR between 5 March 12 UTC and 6

March 12 UTC (Browning, 1986; Dettinger et al., 2015). At 6 March 6 UTC (Fig. S3g), the cold front lies along the densely packed isotherms between the coast of California and 32°N and 124°W, and the warm front is located parallel to the coast of California. This placed the region of warm air advection and the WCB in the southern region of the domain between the two fronts where $\theta_E$ is around 320 K. Water vapor in the AR, which can be sourced from intense vapor transport out of the tropics as well midlatitude convergence of water vapor along the path of the AR, was transported via winds into the WCB (Fig. S3)

(Dettinger et al., 2015). The uplift of the moisture from the AR most likely occurred due to orographic uplift from interaction with complex topography as well as dynamic uplift from the WCB (Fig. S3). It has been suggested that WCBs and ARs can form on their own without direct connection to each other (Dettinger et al., 2015; Dacre et al., 2019). In this case, we observed an AR interacting with a WCB, along with updrafts and hail formation. The synoptic conditions of this event show that the cyclogenesis combined with the dynamical lift of the AR in a convectively unstable environment provided enough updraft to

potentially aid in the electrification of the clouds via hail formation.

### 3.3   Thermodynamic Conditions

Wind in the skew(t) - log(p) diagram at 34.5°N 119.5°W (Fig. 3a) for the time closest to the peak of the event (6 March 6 UTC) indicates strong warm air advection below 800 hPa. This strong veering profile near the surface with increasing wind speeds with height intensifies the mesocyclone and maintains the storm. Most thunderstorms are associated with high values

of Convective Available Potential Energy (CAPE), which measures the amount of energy available for convection. While this storm had values of surface-based CAPE up to 1000 J kg$^{-1}$ as it made its way across the Pacific Ocean toward the west coast of California, there was little to no CAPE in Santa Barbara (10 J kg$^{-1}$) where lightning occurred at 6 March 6 UTC (Fig. 3a, S4). However, like the extreme precipitation events in Cannon et al. (2018), additional dynamical forcing can develop convection even when CAPE is low. Although CAPE was low near Santa Barbara, the proximity of temperature and dew point profiles in

the lower troposphere place the Lifting Condensation Level (LCL) very close to the surface (Fig. 3a).

     Between 800 hPa and 625 hPa, parcels are saturated, indicating the high moisture content of the AR (Fig. 3a, c). The equivalent potential temperature profile (Fig. 3b) shows decreasing $\theta_E$ with increasing height, indicating convective instability at the surface as well as in the midlevels between 800 and 600 hPa. A close inspection of the $\theta_E$ profile at the location of the

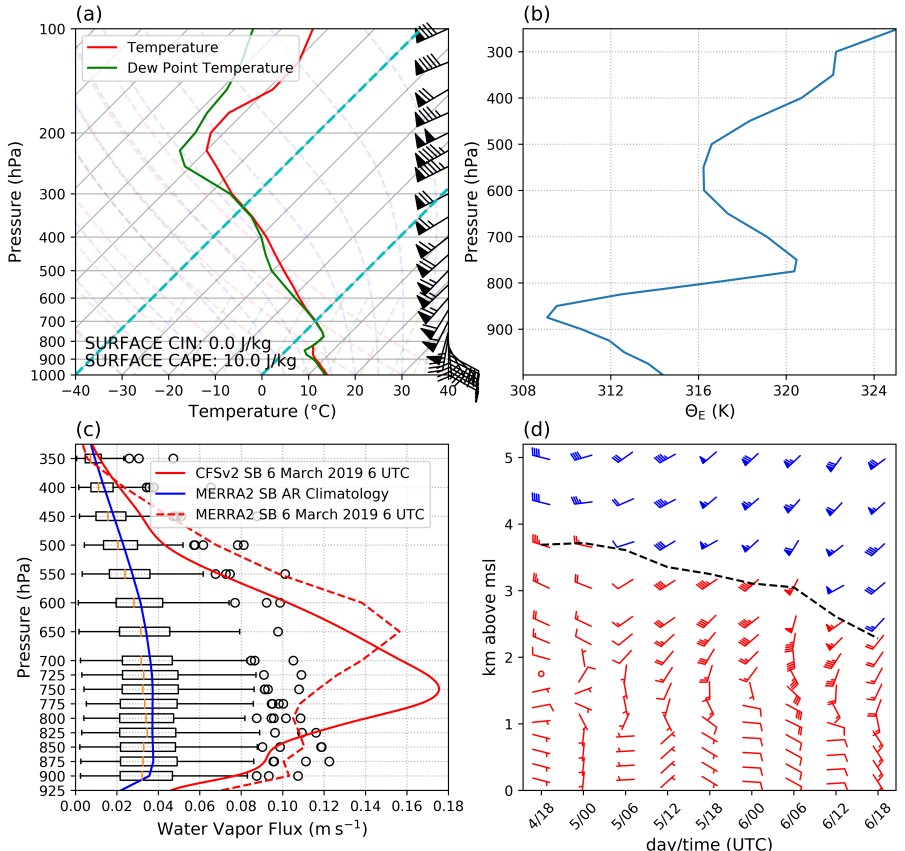

**Figure 3.** (a) Skew(t) - log(p) vertical profile of CFSv2 temperature (red line) and dew point (green line) at 34.5°N and 119.5°W at 6 March 2019 6 UTC. CFSv2 winds (knots; barbs) are indicated on the right side of the figure for each vertical level. Surface values of CAPE and Convective Inhibition (CIN) are shown in the bottom left corner. (b) CFSv2 Equivalent Potential Temperature $\theta_E$ (blue line; K) at 34.5°N and 119.5°W at 6 March 2019 6 UTC. (c) Climatological vertical profile of horizontal water vapor flux (m s$^{-1}$) based on MERRA2 at 34.5°N, 119.375°W for all days when AR conditions are met during the month of March between 1980 and 2015 (i.e. IVT >= 250 kg m$^{-1}$ s$^{-1}$) at this location (blue line and box and whisker plots show the distribution of the 170 events), and vertical profile of horizontal water vapor flux (m s$^{-1}$) based on CFSv2 (red solid line) and MERRA2 (red dashed line) at the same location at 6 March 2019 6 UTC. (d) CFSv2 winds (knots, barbs) at vertical levels (km above mean sea level) at 34.5°N and 119.5°W at 6-hour intervals from 4 March 2019 18 UTC to 6 March 2019 18 UTC. The temperature (°C) is indicated by the color of the barb. Red barbs mean the temperature was greater than 0°C and blue barbs mean the temperature was less than 0°C. The height of the 0°C isotherm is indicated by the black dashed line.

highest lightning flash density (Fig. S4) indicates convective instability at nearly every 6-hour time step for the duration of the storm. The horizontal water vapor flux (m s$^{-1}$) calculated at each pressure level on 6 March 6 UTC at 34.5°N, 119.5°W, (Fig. 3c) indicates that the water vapor flux peaked at 0.17 m s$^{-1}$ between 700 and 800 hPa. Similar results were found when using MERRA2, although with a slightly lower water vapor flux that occurred around 650 hPa (Fig. 3c). Compared to the

climatological vertical profile of water vapor flux from the past March AR events in Santa Barbara, the AR on 6 March 2019 was extremely moist with maximum moisture peaking at a higher than average pressure level (Fig. 3c). The height (km above mean sea level) of the 0°C isotherm at 34.5°N, 119.5°W (Fig. 3d) is around 2.5 km above mean sea level during the peak of the storm, which is below the average height of the 0°C isotherm during past AR events in Santa Barbara (Fig. 2d). The lifting of moist layers in these thermodynamic conditions either orographically or by the WCB resulted in conditionally unstable air and strong updrafts below freezing level (Fig. 3c, 3d). GOES-17 Cloud and Moisture Imagery Brightness Temperature (Fig. S5) indicates vigorous convection via cold cloud temperatures that decrease to approximately -71°C near Santa Barbara at the time closest to the peak of the event. These cold top temperatures indicate a very strong updraft, which would result in hail formation when water droplets in the region of the updraft are carried above the freezing level (Wallace and Hobbs, 2006; Pruppacher and Klett, 1997). The lifting of the moist layers as well as the convective updrafts contributed to the formation of hail with an average size of 13.5 mm, which is co-located with cold cloud top temperatures (Fig. S5), indicating the importance of deep convective updrafts for the development of the thunderstorms.

## 3.4 Lightning Conditions

Convective updraft in the lower troposphere are considered important for the build-up of regions with positive and negative net charges in the mixed-phase region of the cloud (0°C to -40°C), playing a role in the onset of lightning and thunder (Price and Rind, 1993; Price, 2013; Pessi and Businger, 2009; Doswell, 2001). Enhanced updrafts increase electrification and lightning rates because they transport droplets to below freezing levels increasing ice mass (Pessi and Businger, 2009). Downdrafts into the mixed-phase region of the cloud may aid in pushing hailstones downward and are important mechanisms for electrification of the storm (Price and Rind, 1993; Price, 2013). When the updrafted droplets and downdrafted hailstones collide, they can release latent heat, and potentially form graupel, a softer form of hail that is warmer than its environment (Price and Rind, 1993; Doswell, 2001). Particles in the mixed phase region of the cloud can collide with graupel and acquire positive charges when ascending (negative when descending). Over time, this process changes the storm cloud microphysics and electrical charges resulting in a negatively charged base and a positively charged top (Doswell, 2001; Price, 2013).

In the March 2019 storm, updrafts in the deep convective clouds, identified by overshooting cloud tops (Fig. S5), could have transported smaller droplets to above the freezing level, (below 700 hPa), potentially allowing for the formation of hail with a positive charge (Fig. 3a, 3d, 1b, S5). At the time closest to the peak of the event in Santa Barbara, dry air was entrained between 600 hPa and 400 hPa as well as in the upper-levels between 300 hPa and 200 hPa (Fig 3a), which could have enhanced downdrafts contributing in the formation of electrified hailstones.

According to Price and Rind (1993), the proportion of in-cloud (IC) to cloud-to-ground (CG) lightning pulses in thunderstorms is well correlated with the thickness of the cloud region between 0°C and the top of the cloud. Therefore, as the thickness of the thunderstorm cloud increases, the ratio of IC to CG also increases. Here we use cloud-top height from GOES-R (GOES-17) Advanced Baseline Imager Level 2 (Fig. S7) and the height of the 0°C isotherm (Fig. S8) as a proxy for cloud thickness. Figure S6a shows the number of IC pulses and CG pulses at every 15 minutes between 4 March 0 UTC and 7 March 0 UTC in the region of the extratropical cyclone. Between 4 March 0 UTC and 12 UTC, there are between 2000 and 3000 CG pulses

and about 1000 to 2000 IC pulses centered around 26°N and 136°W. The second peak in lightning occurs at approximately 5 March 12 UTC with almost 4000 CG pulses and 3000 IC pulses centered around 30°N and 128°W (Fig. S6a). The last peak of lightning frequency occurred between 6 March 0 UTC and 6 UTC with approximately 3000 IC pulses and less than 1000 CG pulses centered at 34°N and 120°W (Fig. S6a). The cloud top height near the lightning throughout the event is between 9,000 m and 10,000 m (Fig. S7). However, the 0°C isotherm near the lightning drops closer to the ground as time passes, indicating that the cloud thickness increases as the event progresses (Fig. S8). The height of the IC pulses are below 5000 m before 5 March 12 UTC and between 7500 and 10000 m after 5 March 18 UTC (Fig. S6b). The increased IC pulse height (Fig. S6b) could be explained by the increased cloud thickness between the height of 0°C isotherm and the cloud top in the later half of the storm (after 5 March 18 UTC), similar to the findings of Price and Rind (1993).

## 4 Conclusions

On the coast of the Santa Barbara, CA, an extratropical cyclone and an AR made landfall at 5 March 2019 12 UTC. The AR intensified until its peak at 6 March 6 UTC, resulting in precipitation via uplift from the WCB and orographic forcing. This event was associated with cold top clouds and vigorous convection that reached its peak at 6 March 2019 4 UTC. While the accumulated rainfall seen during this storm (about 23 mm) are not uncommon in winter storms associated with ARs making landfall in Southern California, this system exhibited extraordinary lightning activity for the region. In 30 hours between 5 March 12 UTC and 6 March 18 UTC, ENGLN detected 73,442 flashes of lightning with 119,363 combined IC and CG pulses around Southern California (20°N to 50°N and 140°W to 110°W). Of those, 1,486 lightning pulses occurred over Santa Barbara County in the 24 hours following 6 March 0 UTC, 533 of which were cloud-to-ground type.

The lightning activity can be considered highly unusual in a region that observes, on average less than 23 lightning flashes in the entire month of March. Although the system evolved as a typical winter storm associated with a cutoff low, it was exceptional due to the high water vapor content provided by the AR, particularly at mid-levels of the atmosphere. The AR developed in an troposphere cooler than average for an AR, as indicated by the low elevation of the 0°C isotherm (about 2.5 km above mean sea level). The AR provided higher than average horizontal water vapor flux between 800 and 600 hPa compared to other March landfalling ARs in Santa Barbara. Unlike most thunderstorms in the tropics, this event was not characterized by significant CAPE when the storm approaches the coast. However, thermodynamic profiles indicated layers with potential instability near the surface and in the mid-troposphere throughout the life cycle of the thunderstorms. The uplift of saturated parcels in a convectively unstable atmosphere from the WCB and further by the orographic forcing resulted in enhanced updrafts. These updrafts transported droplets in a cold environment and high moisture availability from the AR, providing the ingredients to form hail. Downdrafts enhanced by entrainment between 600 hPa and 400 hPA may have contributed to the downward transport hail, helping to transform the charge distribution in the clouds enhancing lightning activity. Understanding the dynamics of this storm provides the theoretical basis for future, systematic investigation of the relationship between ARs and unusual lightning scenarios in other regions. It also is critical to understand these processes in populated areas such as Santa Barbara, where lightning can significantly increase hazards during rainfall events.

*Code and data availability.* The code for this analysis can be found at https://github.com/dlnash/arthunderstorm2019. May et al. (2008 - 2017) was used for the development of some of the figures. CFSv2 data (Saha et al., 2014, https://www.ncdc.noaa.gov/data-access/ model-data/model-datasets/climate-forecast-system-version2-cfsv2), TRMM LIS-OTD lightning climatology (Cecil, 2015, https://ghrc.nsstc. nasa.gov/uso/ds_details/collections/loCv2.3.2015.html), GOES-R data (GOES-R Algorithm Working Group and GOES-R Series Program, 2017, https://data.nodc.noaa.gov/cgi-bin/iso?id=gov.noaa.ncdc:C01502), MERRA-2 data (Global Modeling and Assimilation Office (GMAO), 2015; Gelaro et al., 2017a, https://disc.gsfc.nasa.gov/datasets/M2I6NPANA_V5.12.4/summary?keywords=MERRA2), and NOAA NEXRAD L3 data (NOAA National Weather Service (NWS) Radar Operations Center, 2019, https://data.nodc.noaa.gov/cgi-bin/iso?id=gov.noaa.ncdc: C00708) are all freely available online. The global AR database based on MERRA-2 and the detection algorithm from Guan and Waliser (2015) used to identify AR events between 1980 and 2019 are freely available at https://ucla.box.com/ARcatalog. The lightning data used for this study was freely provided by Earth Networks (Earth Networks, 2019).

## Appendix A: Calculation of IVT

Integrated water vapor transport (IVT), a variable widely used for the detection and identification of ARs (e.g. (Guan and Waliser, 2015; Ralph et al., 2019; Dettinger et al., 2015)) is derived from specific humidity and wind fields at 17 pressure levels between 1,000 and 300 hPa inclusive from the CFSv2 operational analysis. IVT is calculated in the zonal (x) and meridional (y) direction using the following equations:

$$IVT_x = -\frac{1}{g} \int_{1000}^{300} uq\,dp \tag{A1}$$

$$IVT_y = -\frac{1}{g} \int_{1000}^{300} vq\,dp \tag{A2}$$

where g is the gravitational acceleration, u is zonal wind, v is meridional wind, q is specific humidity, p is pressure, and the column integration is between pressure levels 1000 and 300 hPa inclusive.

*Author contributions.* LC conceptualized this article and participated in the interpretation of the data. DN participated in data collection, analysis, interpretation, and drafting of the article. All authors participated in the revision and final version of the article.

*Competing interests.* The authors declare that they have no conflict of interest.

*Acknowledgements.* This research was supported by NASA Headquarters under the NASA Earth and Space Science Fellowship Program - Grant 80NSSC18K1412. The authors would like to thank Duane Waliser and Bin Guan at NASA's Jet Propulsion Laboratory in Pasadena,

CA, Tessa Montini at University of California, Santa Barbara and Forest Cannon at Scripps Institution of Oceanography for insights into this
event. The authors would also like to thank Mike Eliason of the Santa Barbara County Fire Department for providing the lightning picture in
Fig. 1a.

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
