# Peer review of "Brief Communication: An Electrifying Atmospheric River: Understanding the Thunderstorm Event in Santa Barbara County during March 2019"

_Natural Hazards and Earth System Sciences, 2019_

## Referee Comment (RC1) · Anonymous Referee #1 · 1 Jan 2020

Review of 'Brief Communication: An Electrifying Atmospheric River: Understanding the Thunderstorm Event in Santa Barbara County during March 2019' by Nash and Carvalho.

Overview: This brief manuscript describes meteorological characteristics of an atmospheric river event from March 2019 that caused an unprecedented amount of lightning in Santa Barbara. The manuscript is clearly written, and the analysis is straightforward: brief but appropriate for publication as a brief communication. My biggest concern with the manuscript in its present form is that, while the exceptional nature of the amount of

lightning is well described and detailed, the links between the meteorology and lightning itself are presumed and not very clearly described. The manuscript also restates the lightning results a bit more than necessary given the (very short) length of the manuscript.

Specific comments: L. 7-8, 41-43, and 70-71: This result (i.e., the average flash density for the region) is restated three times by the fourth page of the manuscript. Redundancies such as this example are not warranted in such a brief manuscript, and the text should be tightened up to remove them. The text of the abstract needs particular attention to ensure it conveys the most salient results of the manuscript: I suggest removing this peripheral detail in favor of an additional sentence at the end of the abstract that links the meteorology with the exceptional lightning.

L. 111-112: It's rather difficult to see this synoptic feature (WCB) with such a zoomed-in domain.

L. 147-150 and much of this entire section: Much of this text relays presumptions as conclusions. For example, 'The convective updraft in the lower troposphere was very important for the onset of electrification,...' this manuscript in no way proves what was or wasn't important for the onset of electrification (instead it presents the meteorology, documents that there was quite a lot of electrical activity, and requires inference between the two). This section needs revision to clarify what previous literature suggests are important factors for lots of lightning in storms, and how those factors relate to this particular storm. I was unable to read the citation Price 2013 from the manuscript, but found Pessi and Businger (2009) helpful in framing my review.

L. 193-203: This summary paragraph could do a better job relating what was unusual about this atmospheric river (AR) event that potentially led it to produce so much lightning. ARs in particular are not terribly unusual for Santa Barbara (e.g. Rutz et al. 2014). In addition, the authors suggest that the 2.5km 0 degree C isotherm was a large factor in allowing hail to develop, but 2.5 km is not a particularly low freezing level

for a midlatitude storm at this latitude (Cannon et al. 2017). More care and thought should be put toward this aspect of the manuscript; without this connection the main emphasis of the manuscript becomes a bit fuzzy. The dry air layer at 250 hPa is alluded to as a possible mechanism (and note there is another dry layer at ∼500 hPa).

References Cannon, F., Ralph, F. M., Wilson, A. M., & Lettenmaier, D. P. ( 2017). GPM satellite radar measurements of precipitation and freezing level in atmospheric rivers: Comparison with ground‐based radars and reanalyses. Journal of Geophysical Research: Atmospheres, 122, 12,747– 12,764. https://doi.org/10.1002/2017JD027355

Pessi, A.T. and S. Businger, 2009: Relationships among Lightning, Precipitation, and Hydrometeor Characteristics over the North Pacific Ocean. J. Appl. Meteor. Climatol., 48, 833–848, https://doi.org/10.1175/2008JAMC1817.1

Rutz, J.J., W.J. Steenburgh, and F.M. Ralph, 2014: Climatological Characteristics of Atmospheric Rivers and Their Inland Penetration over the Western United States. Mon. Wea. Rev., 142, 905–921, https://doi.org/10.1175/MWR-D-13-00168.1

---

## Referee Comment (RC2) · Anonymous Referee #2 · 2 Jan 2020

General Comments

This brief communication is appropriate for publication in NHESS. It describes a topical event impacting southern California in March of 2019, and uses remote sensing and operational analysis data sources to better understand the behavior and evolution of this particular potentially hazardous atmospheric river event, which was unusual due to the frequency of lightning strikes.

Specific Comments

Section 1 – what is the purpose of including the information on the peak current – for example, is the strength also an outlier? Also, there are several different numbers used for the flashes over Santa Barbara County in this and other sections (e.g. line 72), please clarify the areas over which these numbers are representing.

Section 2 – consider adding brief justifications for the data sources used. In particular, why GPM for precipitation and not any in situ gauges? How well does GPM estimate precipitation in this region? Please also discuss the implications of using the two different lightning data sources and uncertainties that might result from comparing between the two during different periods.

Section 2 - Consider moving some of the discussion on the lightning observations (e.g. after line 70) into the next section.

Fig 1b - I find the color scale a bit confusing. Consider a scale that goes up only to the maximum of what is in the domain and using a scale that doesn't have the black and brown colors as the highest accumulations.

Line 73 - Consider adding "even" before "if"

Line 117-118, where is this transport from AR to WCB shown?

Line 121 – Is this implying that the combination of the two was necessary for the updrafts, precipitation and hail formation? Perhaps state something more like "In this case, we observed an AR interacting with a WCB, along with updrafts and hail formation" (Please check on other statements of this nature too).

Line 136 – why not just say saturated if the dew point is equal to the temperature?

Figure 3b – I am a little confused on the units. How was this calculated? How is water vapor incorporated?

Section 3.3 - careful with tenses, some examples below in the technical corrections section. Also please make sure it is clear what processes you are hypothesizing played

a role and what you can show played a role based on the data (e.g. paragraph lines 159-166)

Section 4 - Could you explicitly quantify how unusual the lightning is (also in the abstract). It would be helpful to also explicitly quantify the distribution of freezing level – based on prior literature or the datasets you are using here, is this a much colder than normal vertical structure for an AR, or for this area, to make the case for this to be a potential reason behind the high number of lightning strikes?

Technical Corrections

Line 151 – I think "formed" should be "forming" or "allowing the formation of" Line 152 – rephrase to something like "At the time closest to the peak of the event in Santa Barbara, dry air was entrained between 300 hPa and 200 hPa with winds reaching approximately 100 knots (Fig 3a)" Line 160 – should be "warmer than its environment" Line 174 – consider rephrasing to "The last peak of lightning frequency" or something like that

---

## Referee Comment (RC3) · Anonymous Referee #3 · 6 Jan 2020

Review: An Electrifying Atmospheric River: Understanding the Thunderstorm Event in Santa Barbara County during March 2019

The manuscript analyzes an AR event that occurred on March 2019. Despite having a relatively low amount of precipitation (77.6mm in 30h) it was extraordinary amount of lightning strikes in the region that stands out. The topic of this study is of interest to be published and the manuscripts is in general well written however I have a few structural issues that need to deal with before the manuscript is ready for publication.

[Figure]

I believe the entire Introduction section needs to be re-written. In the present form it's a mixture between introduction and results. Therefore, should have the following in mind when re-written the introduction: - What is an ARs; - Possible impacts and benefits (first paragraph should be in here); - Lightening brief introduction and precipitation measures and radar. - I would remove everything that is results from this specific event.

Section 2. Parts of the introduction are already stated here. So, I would keep all the dataset and methodologies in this section and avoid repetition with the introduction.

Section 3. I would include a new sub-section before sub-section "Extratropical Cyclone and AR Conditions". This new sub-section would be the description of the March 2019 event, with most of the information being taken what was already mentioned in the introduction.

Minor comments: Figure 1. The color scales are a bit confusing.

How much do you trust in the vertical speed from reanalysis data?

---

## Author Comment (AC1) · 2 Mar 2020

Dear Reviewer 1,

We appreciate your taking the time to read and comment on our manuscript and thank you for providing constructive criticism and suggestions for improvement. We have uploaded our initial replies to your comments with updated and new figures as a supplement.

On behalf of the authors,

[Figure]

Deanna Nash

Please also note the supplement to this comment:
https://www.nat-hazards-earth-syst-sci-discuss.net/nhess-2019-342/nhess-2019-342-AC1-supplement.pdf

**Supplement:**

**Author response to Anonymous Referee 1 for "Brief Communication: An Electrifying Atmospheric River: Understanding the Thunderstorm Event in Santa Barbara County during March 2019" by Deanna Nash and Leila M.V. Carvalho.**

Responses to reviewer comments are given in blue text. New or changed text is given in italics (bold italics for emphasis where
5  noted)
* * *
**General Comments**

This brief manuscript describes meteorological characteristics of an atmospheric river event from March 2019 that caused an unprecedented amount of lightning in Santa Barbara. The manuscript is clearly written, and the analysis is straightforward:
10  brief but appropriate for publication as a brief communication. My biggest concern with the manuscript in its present form is that, while the exceptional nature of the amount of lightning is well described and detailed, the links between the meteorology and lightning itself are presumed and not very clearly described. The manuscript also restates the lightning results a bit more than necessary given the (very short) length of the Manuscript.

We thank the reviewer for the time taken to review this manuscript and all constructive feedback that helped improve the
15  paper, specifically in regard to clarifying the links between the lightning and thermodynamics. Below we address the specific comments.

**Specific Comments**

L. 7-8, 41-43, and 70-71: This result (i.e., the average flash density for the region) is restated three times by the fourth page of the manuscript. Redundancies such as this example are not warranted in such a brief manuscript, and the text should be
20  tightened up to remove them. The text of the abstract needs particular attention to ensure it conveys the most salient results of the manuscript: I suggest removing this peripheral detail in favor of an additional sentence at the end of the abstract that links the meteorology with the exceptional lightning.

We agree with the reviewer and have updated these sections to remove the redundancies. The final sentence of the abstract now states, *"Despite the negligible convective available potential energy (CAPE) during the peak of the thunderstorm near*
25  *Santa Barbara, the lifting of layers with high water vapor content in the AR via warm conveyor belt and orographic forcing in a convectively unstable atmosphere resulted in the formation of hail and enhanced electrification."*

L. 111-112: It's rather difficult to see this synoptic feature (WCB) with such a zoomed-in domain.

We agree it was difficult to see the WCB in the previous version of the figure. We have zoomed out on the domain of Fig. S3 and have added contours of IVT to the maps for reference to the location of the AR in relation to the WCB (noted by RC2).
30  See below for updated Fig. S3

L. 147-150 and much of this entire section: Much of this text relays presumptions as conclusions. For example, 'The convective updraft in the lower troposphere was very important for the onset of electrification,...' this manuscript in no way proves what was or wasn't important for the onset of electrification (instead it presents the meteorology, documents that there was quite a lot of electrical activity, and requires inference between the two). This section needs revision to clarify what previous
35  literature suggests are important factors for lots of lightning in storms, and how those factors relate to this particular storm. I was unable to read the citation Price 2013 from the manuscript, but found Pessi and Businger (2009) helpful in framing my review.

We agree that clarification was needed between what previous literature suggests are important factors in electrification and what the data implies about the electrification for this particular storm. We thank the reviewer for suggesting the reference Pessi
40  and Businger (2009) which was added to this manuscript. The section on lightning conditions in the results has been edited extensively for these clarifications.

L. 193-203: This summary paragraph could do a better job relating what was unusual about this atmospheric river (AR) event that potentially led it to produce so much lightning. ARs in particular are not terribly unusual for Santa Barbara (e.g. Rutz et al. 2014). In addition, the authors suggest that the 2.5km 0 degree C isotherm was a large factor in allowing hail to develop, but 2.5
45  km is not a particularly low freezing level for a midlatitude storm at this latitude (Cannon et al. 2017). More care and thought

should be put toward this aspect of the manuscript; without this connection the main emphasis of the manuscript becomes a bit fuzzy. The dry air layer at 250 hPa is alluded to as a possible mechanism (and note there is another dry layer at 500 hPa).

    According to Rutz et al. (2014), AR frequency in the Santa Barbara coastal region is approximately 6% of the time steps in ERA-Interim analyses (see Rutz et al. (2014) Fig 4a) between November 1988 to April 2011 (Nov-April only), meaning that an AR was identified at approximately 1,000 6-hour time steps or roughly 250 AR days out of 4,100. The global AR detection algorithm developed by Guan and Waliser (2015) used to identify previous AR days in our study shows similar agreement to the results of Rutz et al. (2014). We think that 6% of the time is a relatively infrequent occurrence for ARs, and Southern California has the lowest frequency of ARs compared to other regions along the west coast of North America (Harris and Carvalho, 2018; Guan and Waliser, 2015, among others) However, our manuscript shows that despite the AR having a relatively average IVT value for ARs that make landfall in Santa Barbara (around 400 kg m$^{-1}$ s$^{-1}$; see Fig. S2b), when looking at the vertical profile of the horizontal water vapor flux at each pressure level, the AR that occurred on 5-6 March 2019 had a significantly above average water vapor flux content in the middle troposphere compared to the other 170 days and AR made landfall in Santa Barbara during the month of March (see Fig. 3c). To show what was unique and what was not for this particular AR, we have updated Fig. S2 to include distribution information of the characteristics of ARs in Santa Barbara, including 0°C Isotherm Height.

    According to Cannon et al. (2017), the mean height of the 0°C isotherm was about 2,500 m for the 83 AR events in Central and Northern California (20°N to 60°N and 160°W to 110°W) during three winter seasons (October through March; 2014-2017). After calculating the height of the 0°C isotherm using MERRA2 and the methodology used in Cannon et al. (2017) and Harris Jr et al. (2000), we created a climatology of the 0°C isotherm for all days an AR made landfall in Santa Barbara (identified using the AR detection algorithm provided by Guan and Waliser (2015)) between 1980 and 2017 (n=1814), and found that the average height of the 0°C isotherm during these days was about 3500 m (Fig. S2d). Therefore, the 0°C isotherm during the 5 March AR was below average for the location and the period.

    We do agree with the reviewer that the connection between the meteorological conditions and the lightning could be improved to demonstrate the reasons for the unusual lightning strikes. To properly address these issues we included new results regarding the profile of convective (potential) instability in the location of the highest lightning flash density (see new Fig. S4), identified by the profiles of Equivalent Potential Temperature ($\theta_E$). We have also added the $\theta_E$ profile to Fig. 3 to indicate the importance of convective instability in Santa Barbara at the time of the peak of the event (see Fig. 3b). With this information we provided additional evidence that the deep moist atmosphere lifted via WCB and orographic forcing in a convectively unstable atmosphere with a low 0°C isotherm was highly conducive to hail formation and lightning, even under conditions of relatively low CAPE.

[Figure]

**Figure 1. Figure S3.** CFSv2 850 hPa Equivalent Potential Temperature (shaded; K), 850 hPa winds (barbs; knots), and IVT greater than 250 kg m$^{-1}$ s$^{-1}$ (white contours; every 250 kg m$^{-1}$ s$^{-1}$) for each 6-hour time step between 4 March 2019 18 UTC and 6 March 2019 18 UTC.

[Figure]

**Figure 2. Figure S2.** (a) CFSv2 IVT (red line; kg m$^{-1}$ s$^{-1}$) and IWV (blue line; mm) at the grid cell closest to Santa Barbara (34.5°N, 119.5°W) at each 6-hour time step between 4 March 2019 18 UTC and 6 March 2019 18 UTC. The minimum thresholds for the location to be considered part of an AR event are indicated by the dotted lines. (b) Mean IVT of the AR objects that made landfall in Santa Barbara in all the months (blue lines) and only March (grey lines) between January 1980 and May 2019 based on the AR Catalog from Guan and Waliser (2015). The mean IVT for the AR Event on March 5 is shown by the red solid line. The means of the distributions are shown in the dotted line. (c) Same as (b) but for direction of mean IVT propagation (azimuth is 0° if IVT is directed to the north). (d) Same as (b) but for the height of the 0°C Isotherm (m) interpolated from MERRA2 temperature and geopotential height.

[Figure]

**Figure 3. Figure S4.** *(left panel)* Skew(t) - log(p) vertical profile of CFSv2 temperature (red line) and dew point (green line) at the grid cell with the highest flash density (per 6 hours); *(right top panel)* CFSv2 CAPE (shaded, J kg⁻¹) and MSLP (black dashed contours; hPa) with the location of the highest flash density indicated by the red dot; *(right bottom panel)* CFSv2 Equivalent Potential Temperature (blue line; K) at the grid cell with the highest flash density for each 6-hour time step between (a) 4 March 2019 18 UTC and (i) 6 March 2019 18 UTC.

[Figure]

**Figure 4. Figure 3.** (a) Skew(t) - log(p) vertical profile of CFSv2 temperature (red line) and dew point (green line) at 34.5°N and 119.5°W at 6 March 2019 06 UTC. CFSv2 winds (knots; barbs) are indicated on the right side of the figure for each vertical level. Surface values of CAPE and Convective Inhibition (CIN) are shown in the bottom left corner. (b) CFSv2 Equivalent Potential Temperature $\theta_E$ (blue line; K) at 34.5°N and 119.5°W at 6 March 2019 06 UTC. (c) Climatological vertical profile of horizontal water vapor flux (m s$^{-1}$) based on MERRA2 at 34.5°N, 119.375°W for all days when AR conditions are met during the month of March between 1980 and 2015 (i.e. IVT >= 250 kg m$^{-1}$ s$^{-1}$) at this location (blue line and box and whisker plots show the distribution of the 170 events), and vertical profile of horizontal water vapor flux (m s$^{-1}$) based on CFSv2 (red solid line) and MERRA2 (red dashed line) at the same location at 6 March 2019 06 UTC. (d) CFSv2 winds (knots, barbs) at vertical levels (km above mean sea level) at 34.5°N and 119.5°W at 6-hour intervals from 4 March 2019 18 UTC to 6 March 2019 18 UTC. The temperature (°C) is indicated by the color of the barb. Red barbs mean the temperature was greater than 0°C and blue barbs mean the temperature was less than 0°C. The height of the 0°C isotherm is indicated by the black dashed line.

**References**

Cannon, F., Ralph, F. M., Wilson, A. M., and Lettenmaier, D. P.: GPM Satellite Radar Measurements of Precipitation and Freezing Level in Atmospheric Rivers: Comparison With Ground-Based Radars and Reanalyses, Journal of Geophysical Research: Atmospheres, 122, 12–747, https://doi.org/https://doi.org/10.1002/2017JD027355, 2017.

80 Guan, B. and Waliser, D.: Detection of atmospheric rivers: Evaluation and application of an algorithm for global studies, Journal of Geophysical Research: Atmospheres, 120, 12 514–12 535, https://doi.org/10.1002/2015JD024257, 2015.

Harris, S. M. and Carvalho, L. M.: Characteristics of southern California atmospheric rivers, Theoretical and applied climatology, 132, 965–981, https://doi.org/10.1007/s00704-017-2138-1, 2018.

Harris Jr, G. N., Bowman, K. P., and Shin, D.-B.: Comparison of freezing-level altitudes from the NCEP reanalysis with

85 TRMM precipitation radar brightband data, Journal of climate, 13, 4137–4148, https://doi.org/https://doi.org/10.1175/1520-0442(2000)013<4137:COFLAF>2.0.CO;2, 2000.

Pessi, A. T. and Businger, S.: Relationships among lightning, precipitation, and hydrometeor characteristics over the North Pacific Ocean, Journal of Applied Meteorology and Climatology, 48, 833–848, https://doi.org/10.1175/2008JAMC1817.1, 2009.

Rutz, J. J., Steenburgh, W. J., and Ralph, F. M.: Climatological characteristics of atmospheric rivers and their inland penetration over the

90 western United States, Monthly Weather Review, 142, 905–921, https://doi.org/https://doi.org/10.1175/MWR-D-13-00168.1, 2014.

---

## Author Comment (AC2) · 2 Mar 2020

Dear Reviewer 2,

We appreciate your taking the time to read and comment on our manuscript and thank you for providing constructive criticism and suggestions for improvement. We have uploaded our initial replies to your comments with updated and new figures as a supplement.

On behalf of the authors,

[Figure]

Deanna Nash

Please also note the supplement to this comment:
https://www.nat-hazards-earth-syst-sci-discuss.net/nhess-2019-342/nhess-2019-342-AC2-supplement.pdf
* * *
[Figure]

**Supplement:**

**Author response to Anonymous Referee 2 for "Brief Communication: An Electrifying Atmospheric River: Understanding the Thunderstorm Event in Santa Barbara County during March 2019" by Deanna Nash and Leila M.V. Carvalho.**

Responses to reviewer comments are given in blue text. New or changed text is given in italics (bold italics for emphasis where noted)
* * *
**General Comments**

This brief communication is appropriate for publication in NHESS. It describes a topical event impacting southern California in March of 2019, and uses remote sensing and operational analysis data sources to better understand the behavior and evolution of this particular potentially hazardous atmospheric river event, which was unusual due to the frequency of lightning strikes.

We thank the reviewer for the time spent to review this manuscript and all constructive feedback that helped improve the paper. Please see responses to comments below.

**Specific Comments**

Section 1 – what is the purpose of including the information on the peak current – for example, is the strength also an outlier?

The information on the peak current was included since this was the first documented case of such an extreme number of lightning strikes occurring in this region; we believe it is important to include statistics of these lightning flashes for future comparisons. Since it is not important to the main message of this manuscript, we have removed it for brevity.

Also, there are several different numbers used for the flashes over Santa Barbara County in this and other sections (e.g. line 72), please clarify the areas over which these numbers are representing.

We made an effort to clarify the areas over which flash densities have occurred, for example, line 72 now reads, *"TRMM LIS-OTD records an area climatological average of 9.15 flashes per day in the region surrounding southern California (20°N to 50°N and 140°W to 110°W), making the 14,416 lightning flashes in under 24 hours very extreme. In fact, even if this was the only lightning activity for 2019, it would represent about 1,500 times the climatological rate (Fig. 1d)."*

Section 2 – consider adding brief justifications for the data sources used. In particular, why GPM for precipitation and not any in situ gauges? How well does GPM estimate precipitation in this region?.

We have done some preliminary analysis into two additional different precipitation products, namely three in-situ gauges in the SB region from National Weather Service and radar data from NWS (see Figure 1 below - not included in manuscript). The plot below shows the precipitation accumulation for the duration of the storm for the different locations and data sources. The gauges and the radar seem to have similar values for precipitation, except in the last location, while GPM values fall above the gauge and radar measurements in all three locations. The authors have decided to switch the precipitation dataset to NOAA NEXRAD L3 precipitation accumulation estimates (which are similar the the rain gauges) for this manuscript. Since there was not a significant amount of precipitation during this particular event (less than 30 mm accumulated), and the main point of this article focuses on the lightning, we use NOAA NEXRAD L3 precipitation to show that there was in fact precipitation to support our conclusion that the hail identified by NOAA NEXRAD L3 was important for the electrification process.

[Figure]

**Figure 1.** Precipitation accumulation (mm) over the duration of the storm period for 3 different locations near Santa Barbara for 3 different data sources, including NWS precipitation gauges (blue line), NOAA NEXRAD L3 precipitation estimates (orange line), and IMERG GPM (green line).

35  Please also discuss the implications of using the two different lightning data sources and uncertainties that might result from comparing between the two during different periods.

A few sentences have been added to the Data and Methods section describing how results could be impacted by two different lightning data sources. *"Comparing the two lightning sources has a certain level of uncertainty, due to the fact that TRMM LIS-OTD and ENGLN do not overlap temporally. However, because this event had significantly above average lightning flash*
40  *rates compared to the climatology, the possible error introduced by comparing two different data sets does not impact the results."*

Section 2 - Consider moving some of the discussion on the lightning observations (e.g. after line 70) into the next section.

We have moved the discussion on the lightning observations to the lightning results section.

Fig 1b - I find the color scale a bit confusing. Consider a scale that goes up only to the maximum of what is in the domain
45  and using a scale that doesn't have the black and brown colors as the highest accumulations.

Colormaps in Fig. 1 were updated to only go to the maximum, and black and brown were removed from the colormaps. See the updated figure below.

[Figure]

**Figure 2. Figure 1.** (a) Photo of lightning at the Santa Barbara Harbor in Santa Barbara, CA taken by Mike Eliason from the Santa Barbara County Fire Department during the storm at 6 March 2019 04 UTC. (b) NOAA NEXRAD L3 precipitation accumulation (shaded; mm) and locations of NOAA NEXRAD L3 Hail Signatures (black points) between 5 March 2019 12 UTC and 6 March 2019 23:59 UTC. The location of Santa Barbara is indicated by the white star. (c) ENGLN lightning strike frequency (shaded; flashes day$^{-1}$) on 6 March 2019. The location of Santa Barbara is indicated by the black star. (d) Climatological mean lightning density (shaded; flashes day$^{-1}$) between 1995 and 2014 using TRMM LIS-OTD lightning climatology and lightning strike locations (red points) between 04 and 05 UTC on 6 March 2019 based on ENGLN. The location of Santa Barbara is indicated by the black star.

Line 73 - Consider adding "even" before "if"

We added "even" before "if" on line 73, so the sentence now reads, "In fact, *even* if this was the only lightning activity for 2019, it would represent about 1,500 times the climatological rate."

Line 117-118, where is this transport from AR to WCB shown?

The paragraph has been updated to say that both the WCB and AR can be seen in Figure S3. Fig. S3, which has been updated as per RC1's comment, has also been updated to show the location of the AR in relation to the WCB.

[Figure]

**Figure 3. Figure S3.** CFSv2 850 hPa Equivalent Potential Temperature (shaded; K), 850 hPa winds (barbs; knots), and IVT greater than 250 kg m$^{-1}$ s$^{-1}$ (white contours; every 250 kg m$^{-1}$ s$^{-1}$) for each 6-hour time step between 4 March 2019 18 UTC and 6 March 2019 18 UTC.

Line 121 – Is this implying that the combination of the two was necessary for the updrafts, precipitation and hail formation? Perhaps state something more like "In this case, we observed an AR interacting with a WCB, along with updrafts and hail formation" (Please check on other statements of this nature too).

The wording on this statement and other statements like this have been updated to clarify that there is a connection rather than imply that one was necessary for the other.

Line 136 – why not just say saturated if the dew point is equal to the temperature?

60     The sentence has been updated to now says, "Between 800 hPa and 625 hPa, *parcels are saturated*, indicating the high moisture from the AR (Fig. 3a, b)."

    Figure 3b – I am a little confused on the units. How was this calculated? How is water vapor incorporated?

    The vertical profile of horizontal water vapor fluxes (m s-1) are the fluxes at each pressure level. At each pressure level, water vapor flux in the v direction is calculated by multiplying the v component wind and specific humidity (q) (same for u

65 direction, but with u component wind). Then the magnitude is calculated by taking the square root of the v flux squared plus the u flux squared.

$$VT_u(ms^{-1}) = q(kgkg^{-1}) * u(ms^{-1}) \tag{1}$$

$$VT_v(ms^{-1}) = q(kgkg^{-1}) * v(ms^{-1}) \tag{2}$$

70

$$VT = \sqrt{VT_u^2 + VT_v^2} \tag{3}$$

Since specific humidity is unitless (kg kg$^{-1}$), it takes on the units of the wind component (m s$^{-1}$). Figure 8b in Guan and Waliser (2015) uses the mean vertical profiles of horizontal water vapor fluxes (m s$^{-1}$) to highlight the low-level nature of ARs across different regions. This figure allows us to show where the moisture of this particular AR is focused, as well as point out the

75 above-average moisture levels compared to other landfalling ARs in the Santa Barbara region.

    Section 3.3 - careful with tenses, some examples below in the technical corrections section. Also please make sure it is clear what processes you are hypothesizing played a role and what you can show played a role based on the data (e.g. paragraph lines 159-166)

    We have updated the tenses using the examples in the technical corrections section. We have also clarified throughout

80 section 3.3 the processes we are hypothesizing played a role in the electrification based on previous literature and where the connections are between that and what the data shows.

    Section 4 - Could you explicitly quantify how unusual the lightning is (also in the abstract).

    The conclusions section has been updated to quantify how unusual the lightning is, stating, *"In 30 hours between 5 March 12 UTC and 6 March 18 UTC, ENGLN detected 73,442 flashes of lightning with 119,363 combined in-cloud (IC) and cloud-*

85 *to-ground (CG) pulses around Southern California (20°N to 50°N and 140°W to 110°W). Of those, 1,486 lightning pulses occurred over Santa Barbara County in the 24 hours following 6 March 2019 00 UTC, 533 of which were cloud-to-ground type. The lightning activity can be considered highly unusual for this region that observes, on average, less than 10 flashes per day."* We have also added a statement to the abstract that states, *"The Earth Networks Global Lightning Network (ENGLN) detected 14,416 lightning flashes in southern California (20°N to 50°N and 140°W to 110°W) in 24 hours, which is roughly*

90 *1500 times the climatological flash rate in this region."* Additionally, in our results, we state, *"TRMM LIS-OTD records an area climatological average of 9.15 flashes per day in the region surrounding southern California (20°N to 50°N and 140°W to 110°W), making the 14,416 lightning flashes in under 24 hours very extreme. In fact, even if this was the only lightning activity for 2019, it would represent about 1,500 times the climatological rate (Fig. 1d) (Cecil, 2015)."*

    It would be helpful to also explicitly quantify the distribution of freezing level – based on prior literature or the datasets you

95 are using here, is this a much colder than normal vertical structure for an AR, or for this area, to make the case for this to be a potential reason behind the high number of lightning strikes?

    A similar comment from RC1 was made. According to Cannon et al. (2017), the mean height of the 0°C isotherm was about 2,500 m for the 83 AR events in Central and Northern California (20°N to 60°N and 160°W to 110°W) during three winter seasons (October through March; 2014-2017). We have decided to calculate the height of the 0°C isotherm for AR events in SB

100 and add to the supplemental results (see new Fig. S2 below). After calculating the height of the 0°C isotherm using MERRA2 and the methodology used in Cannon et al. (2017) and Harris Jr et al. (2000), we created a climatology of the 0°C isotherm for all days an AR made landfall in Santa Barbara (identified using the AR detection algorithm provided by Guan and Waliser (2015)) between 1980 and 2017 (n=1814), and found that the average height of the 0°C isotherm during these days was about 3500 m (Fig. S2d). This puts the height of the 0°C isotherm for the event below the average.

[Figure]

**Figure 4. Figure S2.** (a) CFSv2 IVT (red line; kg m$^{-1}$ s$^{-1}$) and IWV (blue line; mm) at the grid cell closest to Santa Barbara (34.5°N, 119.5°W) at each 6-hour time step between 4 March 2019 18 UTC and 6 March 2019 18 UTC. The minimum thresholds for the location to be considered part of an AR event are indicated by the dotted lines. (b) Mean IVT of the AR objects that made landfall in Santa Barbara in all the months (blue lines) and only March (grey lines) between January 1980 and May 2019 based on the AR Catalog from Guan and Waliser (2015). The mean IVT for the AR Event on March 5 is shown by the red solid line. The means of the distributions are shown in the dotted line. (c) Same as (b) but for direction of mean IVT propagation (azimuth is 0° if IVT is directed to the north). (d) Same as (b) but for the height of the 0°C Isotherm (m) interpolated from MERRA2 temperature and geopotential height.

**Technical Corrections**

Line 151 – I think "formed" should be "forming" or "allowing the formation of"
  The sentence now reads, *"In the March 2019 storm, updrafts in the deep convective clouds, identified by overshooting cloud tops (Fig. S6), could have transported smaller droplets to above the freezing level, (below 700 hPa), potentially allowing for the formation of hail with a positive charge (Fig. 3c, 1b)."*

Line 152 – rephrase to something like "At the time closest to the peak of the event in Santa Barbara, dry air was entrained between 300 hPa and 200 hPa with winds reaching approximately 100 knots (Fig 3a)"
  The sentence now reads, *"At the time closest to the peak of the event in Santa Barbara, dry air was entrained between 600 hPa and 400 hPa as well as in the upper-levels between 300 hPa and 200 hPa (Fig 3a), which could have enhanced downdrafts contributing in the formation of electrified hailstones."*

115    Line 160 – should be "warmer than its environment"

The sentence now reads, *"When the updrafted droplets and downdrafted hailstones collide, they can release latent heat, and potentially form graupel, a softer form of hail that is warmer than its environment."*

Line 174 – consider rephrasing to "The last peak of lightning frequency" or something like that

The sentence now reads, *"The last peak of lightning frequency occurred between 6 March 00 UTC and 06 UTC with*

120   *approximately 3000 IC pulses and less than 1000 CG pulses centered at 34°N and 120°W (Fig. S7a)."*

**References**

Cannon, F., Ralph, F. M., Wilson, A. M., and Lettenmaier, D. P.: GPM Satellite Radar Measurements of Precipitation and Freezing Level in Atmospheric Rivers: Comparison With Ground-Based Radars and Reanalyses, Journal of Geophysical Research: Atmospheres, 122, 12–747, https://doi.org/https://doi.org/10.1002/2017JD027355, 2017.

Guan, B. and Waliser, D.: Detection of atmospheric rivers: Evaluation and application of an algorithm for global studies, Journal of Geophysical Research: Atmospheres, 120, 12 514–12 535, https://doi.org/10.1002/2015JD024257, 2015.

Harris Jr, G. N., Bowman, K. P., and Shin, D.-B.: Comparison of freezing-level altitudes from the NCEP reanalysis with TRMM precipitation radar brightband data, Journal of climate, 13, 4137–4148, https://doi.org/https://doi.org/10.1175/1520-0442(2000)013<4137:COFLAF>2.0.CO;2, 2000.

125

---

## Author Comment (AC3) · 2 Mar 2020

Dear Reviewer 3,

We appreciate your taking the time to read and comment on our manuscript and thank you for providing constructive criticism and suggestions for improvement. We have uploaded our initial replies to your comments with updated figures as a supplement.

On behalf of the authors,

[Figure]

Deanna Nash

Please also note the supplement to this comment:
https://www.nat-hazards-earth-syst-sci-discuss.net/nhess-2019-342/nhess-2019-342-AC3-supplement.pdf
* * *
[Figure]

**Supplement:**

**Author response to Anonymous Referee 3 for "Brief Communication: An Electrifying Atmospheric River: Understanding the Thunderstorm Event in Santa Barbara County during March 2019" by Deanna Nash and Leila M.V. Carvalho.**

Responses to reviewer comments are given in blue text. New or changed text is given in italics (bold italics for emphasis where
5  noted)
* * *
**General Comments**

The manuscript analyzes an AR event that occurred on March 2019. Despite having a relatively low amount of precipitation
(77.6mm in 30h) it was extraordinary amount of lightning strikes in the region that stands out. The topic of this study is of
10  interest to be published and the manuscripts is in general well written however I have a few structural issues that need to deal
with before the manuscript is ready for publication.
    We thank the reviewer for the time they took to review this paper and the constructive feedback that helped improve the
paper, specifically with regard to the structure of the manuscript. Please see responses to comments below.

**Specific Comments**

15  I believe the entire Introduction section needs to be re-written. In the present form it's a mixture between introduction and
results. Therefore, should have the following in mind when re-written the introduction: - What is an ARs; - Possible impacts
and benefits (first paragraph should be in here); - Lightening brief introduction and precipitation measures and radar. - I would
remove everything that is results from this specific event.
    We have separated the introduction and results, and updated the introduction section to include a paragraph on the back-
20  ground of ARs and their relevance and impact to Southern California. We have also added a short background on lightning to
the introduction.
    Section 2. Parts of the introduction are already stated here. So, I would keep all the dataset and methodologies in this section
and avoid repetition with the introduction.
    Repetition with the introduction has been removed and care has been taken to make sure this section is only dataset and
25  methodologies and no results.
    Section 3. I would include a new sub-section before sub-section "Extratropical Cyclone and AR Conditions". This new
sub-section would be the description of the March 2019 event, with most of the information being taken what was already
mentioned in the Introduction.
    After restructuring the introduction (see above), we have moved the description of the March 2019 event to a new subsection
30  in the results titled "Extratropical Cyclone and AR Conditions" to make sure the results are all in the results section.

**Minor Comments**

Figure 1. The color scales are a bit confusing.
    A similar comment from RC2 was made. The colormaps in Fig. 1 were updated to only go to the maximum, and black and
brown were removed from the colormaps.

[Figure]

**Figure 1. Figure 1.** (a) Photo of lightning at the Santa Barbara Harbor in Santa Barbara, CA taken by Mike Eliason from the Santa Barbara County Fire Department during the storm at 6 March 2019 04 UTC. (b) NOAA NEXRAD L3 precipitation accumulation (shaded; mm) and locations of NOAA NEXRAD L3 Hail Signatures (black points) between 5 March 2019 12 UTC and 6 March 2019 23:59 UTC. The location of Santa Barbara is indicated by the white star. (c) ENGLN lightning strike frequency (shaded; flashes day$^{-1}$) on 6 March 2019. The location of Santa Barbara is indicated by the black star. (d) Climatological mean lightning density (shaded; flashes day$^{-1}$) between 1995 and 2014 using TRMM LIS-OTD lightning climatology and lightning strike locations (red points) between 04 and 05 UTC on 6 March 2019 based on ENGLN. The location of Santa Barbara is indicated by the black star.

35  How much do you trust in the vertical speed from reanalysis data?

We recognize that vertical velocity from reanalysis data is a calculated value. We found that the details provided from vertical velocity in the manuscript were not necessary to show updrafts and deep convection, so we removed vertical velocity from the manuscript and supplement. We have instead decided to focus on what the observed GOES-R infrared brightness temperature tells us, which is that there was an overshooting cloud top at approximately 4:30 UTC, indicating deep convection. We have 40 updated Fig. S5 to highlight this information (see below).

[Figure]

**Figure 2. Figure S5.** Infrared brightness temperatures (shaded, °C) derived from band 13 of the GOES17 ABI L2 Cloud and Moisture Imagery Brightness Temperature at 6 March 2019 4:24 UTC. Detailed infrared brightness temperatures around Santa Barbara (outlined in red) are shown in the top left area of the map.

---

## Author Response (AR1)

**Author response to Anonymous Referee 1 for "Brief Communication: An Electrifying Atmospheric River: Understanding the Thunderstorm Event in Santa Barbara County during March 2019" by Deanna Nash and Leila M.V. Carvalho.**

Responses to reviewer comments are given in blue text. New or changed text is given in italics (bold italics for emphasis where
5  noted)
* * *
**General Comments**

This brief manuscript describes meteorological characteristics of an atmospheric river event from March 2019 that caused an unprecedented amount of lightning in Santa Barbara. The manuscript is clearly written, and the analysis is straightforward:
10  brief but appropriate for publication as a brief communication. My biggest concern with the manuscript in its present form is that, while the exceptional nature of the amount of lightning is well described and detailed, the links between the meteorology and lightning itself are presumed and not very clearly described. The manuscript also restates the lightning results a bit more than necessary given the (very short) length of the Manuscript.

We thank the reviewer for the time taken to review this manuscript and all constructive feedback that helped improve the
15  paper, specifically in regard to clarifying the links between the lightning and thermodynamics. Below we address the specific comments.

**Specific Comments**

L. 7-8, 41-43, and 70-71: This result (i.e., the average flash density for the region) is restated three times by the fourth page of the manuscript. Redundancies such as this example are not warranted in such a brief manuscript, and the text should be
20  tightened up to remove them. The text of the abstract needs particular attention to ensure it conveys the most salient results of the manuscript: I suggest removing this peripheral detail in favor of an additional sentence at the end of the abstract that links the meteorology with the exceptional lightning.

We agree with the reviewer and have updated these sections to remove the redundancies. The final sentence of the abstract now states, *"Despite the negligible convective available potential energy (CAPE) during the peak of the thunderstorm near*
25  *Santa Barbara, the lifting of layers with high water vapor content in the AR via warm conveyor belt and orographic forcing in a convectively unstable atmosphere resulted in the formation of hail and enhanced electrification."*

L. 111-112: It's rather difficult to see this synoptic feature (WCB) with such a zoomed-in domain.

We agree it was difficult to see the WCB in the previous version of the figure. We have zoomed out on the domain of Fig. S5, now Fig. S3, and have added contours of IVT to the maps for reference to the location of the AR in relation to the WCB
30  (noted by RC2).

L. 147-150 and much of this entire section: Much of this text relays presumptions as conclusions. For example, 'The convective updraft in the lower troposphere was very important for the onset of electrification,...' this manuscript in no way proves what was or wasn't important for the onset of electrification (instead it presents the meteorology, documents that there was quite a lot of electrical activity, and requires inference between the two). This section needs revision to clarify what previous
35  literature suggests are important factors for lots of lightning in storms, and how those factors relate to this particular storm. I was unable to read the citation Price 2013 from the manuscript, but found Pessi and Businger (2009) helpful in framing my review.

We agree that clarification was needed between what previous literature suggests are important factors in electrification and what the data implies about the electrification for this particular storm. We thank the reviewer for suggesting the reference Pessi
40  and Businger (2009) which was added to this manuscript. The section on lightning conditions in the results has been edited extensively for these clarifications. Please see updated Section 3.4 Lightning Conditions.

L. 193-203: This summary paragraph could do a better job relating what was unusual about this atmospheric river (AR) event that potentially led it to produce so much lightning. ARs in particular are not terribly unusual for Santa Barbara (e.g. Rutz et al. 2014). In addition, the authors suggest that the 2.5km 0 degree C isotherm was a large factor in allowing hail to develop, but 2.5
45  km is not a particularly low freezing level for a midlatitude storm at this latitude (Cannon et al. 2017). More care and thought

should be put toward this aspect of the manuscript; without this connection the main emphasis of the manuscript becomes a bit fuzzy. The dry air layer at 250 hPa is alluded to as a possible mechanism (and note there is another dry layer at 500 hPa).

According to Rutz et al. (2014), AR frequency in the Santa Barbara coastal region is approximately 6% of the time steps in ERA-Interim analyses (see Rutz et al. (2014) Fig 4a) between November 1988 to April 2011 (Nov-April only), meaning that an AR was identified at approximately 1,000 6-hour time steps or roughly 250 AR days out of 4,100. The global AR detection algorithm developed by Guan and Waliser (2015) used to identify previous AR days in our study shows similar agreement to the results of Rutz et al. (2014). We think that 6% of the time is a relatively infrequent occurrence for ARs, and Southern California has the lowest frequency of ARs compared to other regions along the west coast of North America (Harris and Carvalho, 2018; Guan and Waliser, 2015, among others) However, our manuscript shows that despite the AR having a relatively average IVT value for ARs that make landfall in Santa Barbara (around 400 kg m$^{-1}$ s$^{-1}$; see Fig. S2b), when looking at the vertical profile of the horizontal water vapor flux at each pressure level, the AR that occurred on 5-6 March 2019 had a significantly above average water vapor flux content in the middle troposphere compared to the other 170 days and AR made landfall in Santa Barbara during the month of March (see Fig. 3c). To show what was unique and what was not for this particular AR, we have updated Fig. S3, now Fig. S2 to include distribution information of the characteristics of ARs in Santa Barbara, including 0°C Isotherm Height.

According to Cannon et al. (2017), the mean height of the 0°C isotherm was about 2,500 m for the 83 AR events in Central and Northern California (20°N to 60°N and 160°W to 110°W) during three winter seasons (October through March; 2014-2017). After calculating the height of the 0°C isotherm using MERRA2 and the methodology used in Cannon et al. (2017) and Harris Jr et al. (2000), we created a climatology of the 0°C isotherm for all days an AR made landfall in Santa Barbara (identified using the AR detection algorithm provided by Guan and Waliser (2015)) between 1980 and 2017 (n=1814), and found that the average height of the 0°C isotherm during these days was about 3500 m (Fig. S2d). Therefore, the 0°C isotherm during the 5 March AR was below average for the location and the period.

We do agree with the reviewer that the connection between the meteorological conditions and the lightning could be improved to demonstrate the reasons for the unusual lightning strikes. To properly address these issues we included new results regarding the profile of convective (potential) instability in the location of the highest lightning flash density (see new Fig. S4), identified by the profiles of Equivalent Potential Temperature ($\theta_E$). We have also added the $\theta_E$ profile to Fig. 3 to indicate the importance of convective instability in Santa Barbara at the time of the peak of the event (see Fig. 3b). With this information we provided additional evidence that the deep moist atmosphere lifted via WCB and orographic forcing in a convectively unstable atmosphere with a low 0°C isotherm was highly conducive to hail formation and lightning, even under conditions of relatively low CAPE.

**Author response to Anonymous Referee 2 for "Brief Communication: An Electrifying Atmospheric River: Understanding the Thunderstorm Event in Santa Barbara County during March 2019" by Deanna Nash and Leila M.V. Carvalho.**

Responses to reviewer comments are given in blue text. New or changed text is given in italics (bold italics for emphasis where noted)
* * *
**General Comments**

This brief communication is appropriate for publication in NHESS. It describes a topical event impacting southern California in March of 2019, and uses remote sensing and operational analysis data sources to better understand the behavior and evolution of this particular potentially hazardous atmospheric river event, which was unusual due to the frequency of lightning strikes.

We thank the reviewer for the time spent to review this manuscript and all constructive feedback that helped improve the paper. Please see responses to comments below.

**Specific Comments**

Section 1 – what is the purpose of including the information on the peak current – for example, is the strength also an outlier?

The information on the peak current was included since this was the first documented case of such an extreme number of lightning strikes occurring in this region; we believe it is important to include statistics of these lightning flashes for future comparisons. Since it is not important to the main message of this manuscript, we have removed it for brevity.

Also, there are several different numbers used for the flashes over Santa Barbara County in this and other sections (e.g. line 72), please clarify the areas over which these numbers are representing.

We made an effort to clarify the areas over which flash densities have occurred, for example, this line now reads, *"TRMM LIS-OTD records an area annual average of 9.15 flashes per day in the region surrounding southern California ($20°N$ to $50°N$ and $140°W$ to $110°W$), making the 14,416 lightning flashes in under 24 hours very extreme. In fact, even if this was the only lightning activity for 2019, it would represent about 1,500 times the climatological rate (Fig. 1d)."*

Section 2 – consider adding brief justifications for the data sources used. In particular, why GPM for precipitation and not any in situ gauges? How well does GPM estimate precipitation in this region?.

We have done some preliminary analysis into two additional different precipitation products, namely three in-situ gauges in the SB region from National Weather Service and radar data from NWS (see Figure 1 below - not included in manuscript). The plot below shows the precipitation accumulation for the duration of the storm for the different locations and data sources. The gauges and the radar seem to have similar values for precipitation, except in the last location, while GPM values fall above the gauge and radar measurements in all three locations. The authors have decided to switch the precipitation dataset to NOAA NEXRAD L3 precipitation accumulation estimates (which are similar the the rain gauges) for this manuscript. Since there was not a significant amount of precipitation during this particular event (less than 30 mm accumulated), and the main point of this article focuses on the lightning, we use NOAA NEXRAD L3 precipitation to show that there was in fact precipitation to support our conclusion that the hail identified by NOAA NEXRAD L3 was important for the electrification process.

[Figure]

**Figure 1.** Precipitation accumulation (mm) over the duration of the storm period for 3 different locations near Santa Barbara for 3 different data sources, including NWS precipitation gauges (blue line), NOAA NEXRAD L3 precipitation estimates (orange line), and IMERG GPM (green line).

110    Please also discuss the implications of using the two different lightning data sources and uncertainties that might result from comparing between the two during different periods.

    A few sentences have been added to the Data and Methods section describing how results could be impacted by two different lightning data sources. *"Comparing the two lightning sources has a certain level of uncertainty, due to the fact that TRMM LIS-OTD and ENGLN do not overlap temporally. However, because this event had significantly above average lightning flash*
115  *rates compared to the climatology, the possible error introduced by comparing two different data sets does not impact the results."*

    Section 2 - Consider moving some of the discussion on the lightning observations (e.g. after line 70) into the next section.

    We have moved the discussion on the lightning observations to the lightning results section - now Section 3.4 Lightning Conditions

120    Fig 1b - I find the color scale a bit confusing. Consider a scale that goes up only to the maximum of what is in the domain and using a scale that doesn't have the black and brown colors as the highest accumulations.

    Colormaps in Fig. 1 were updated to only go to the maximum, and black and brown were removed from the colormaps.

    Line 73 - Consider adding "even" before "if"

    We added "even" before "if" on line 73, so the sentence now reads, "In fact, *even* if this was the only lightning activity for
125  2019, it would represent about 1,500 times the climatological rate."

    Line 117-118, where is this transport from AR to WCB shown?

    The paragraph has been updated to say that both the WCB and AR can be seen in Figure S3. Fig. S3, which has been updated as per RC1's comment, has also been updated to show the location of the AR as contour lines in relation to the WCB.

    Line 121 – Is this implying that the combination of the two was necessary for the updrafts, precipitation and hail formation?
130  Perhaps state something more like "In this case, we observed an AR interacting with a WCB, along with updrafts and hail formation" (Please check on other statements of this nature too).

    The wording on this statement and other statements like this have been updated to clarify that there is a connection rather than imply that one was necessary for the other.

    Line 136 – why not just say saturated if the dew point is equal to the temperature?
135    The sentence has been updated to now says, "Between 800 hPa and 625 hPa, *parcels are saturated*, indicating the high moisture from the AR (Fig. 3a, b)."

    Figure 3b – I am a little confused on the units. How was this calculated? How is water vapor incorporated?

    The vertical profile of horizontal water vapor fluxes (m s-1) are the fluxes at each pressure level. At each pressure level, water vapor flux in the v direction is calculated by multiplying the v component wind and specific humidity (q) (same for u
140  direction, but with u component wind). Then the magnitude is calculated by taking the square root of the v flux squared plus the u flux squared.

$$VT_u (m\,s^{-1}) = q(kg\,kg^{-1}) * u(m\,s^{-1}) \qquad\qquad (1)$$

$$VT_v (m\,s^{-1}) = q(kg\,kg^{-1}) * v(m\,s^{-1}) \qquad\qquad (2)$$

145

$$VT = \sqrt{VT_u^2 + VT_v^2} \qquad\qquad (3)$$

Since specific humidity is unitless (kg kg$^{-1}$), it takes on the units of the wind component (m s$^{-1}$). Figure 8b in Guan and Waliser (2015) uses the mean vertical profiles of horizontal water vapor fluxes (m s$^{-1}$) to highlight the low-level nature of ARs across different regions. This figure allows us to show where the moisture of this particular AR is focused, as well as point out the
150  above-average moisture levels compared to other landfalling ARs in the Santa Barbara region.

    Section 3.3 - careful with tenses, some examples below in the technical corrections section. Also please make sure it is clear what processes you are hypothesizing played a role and what you can show played a role based on the data (e.g. paragraph lines 159-166)

We have updated the tenses using the examples in the technical corrections section. We have also clarified throughout section 3.3 the processes we are hypothesizing played a role in the electrification based on previous literature and where the connections are between that and what the data shows.

Section 4 - Could you explicitly quantify how unusual the lightning is (also in the abstract).

The conclusions section has been updated to quantify how unusual the lightning is, stating, *"In 30 hours between 5 March 12 UTC and 6 March 18 UTC, ENGLN detected 73,442 flashes of lightning with 119,363 combined in-cloud (IC) and cloud-to-ground (CG) pulses around Southern California (20°N to 50°N and 140°W to 110°W). Of those, 1,486 lightning pulses occurred over Santa Barbara County in the 24 hours following 6 March 2019 00 UTC, 533 of which were cloud-to-ground type. The lightning activity can be considered highly unusual in a region that observes, on average less than 23 flashes in the entire month of March."* We have also added a statement to the abstract that states, *"The Earth Networks Global Lightning Network (ENGLN) detected 14,416 lightning flashes in southern California (20°N to 50°N and 140°W to 110°W) in 24 hours, which is roughly 1500 times the climatological flash rate in this region."* Additionally, in our results, we state, *"TRMM LIS-OTD records an area annual average of 9.15 flashes per day in the region surrounding southern California (20°N to 50°N and 140°W to 110°W), making the 14,416 lightning flashes in under 24 hours very extreme. In fact, even if this was the only lightning activity for 2019, it would represent about 1,500 times the climatological rate (Fig. 1d) (Cecil, 2015)."*

It would be helpful to also explicitly quantify the distribution of freezing level – based on prior literature or the datasets you are using here, is this a much colder than normal vertical structure for an AR, or for this area, to make the case for this to be a potential reason behind the high number of lightning strikes?

A similar comment from RC1 was made. According to Cannon et al. (2017), the mean height of the 0°C isotherm was about 2,500 m for the 83 AR events in Central and Northern California (20°N to 60°N and 160°W to 110°W) during three winter seasons (October through March; 2014-2017). We have decided to calculate the height of the 0°C isotherm for AR events in Santa Barbara and add to the supplemental results (see new Fig. S2). After calculating the height of the 0°C isotherm using MERRA2 and the methodology used in Cannon et al. (2017) and Harris Jr et al. (2000), we created a climatology of the 0°C isotherm for all days an AR made landfall in Santa Barbara (identified using the AR detection algorithm provided by Guan and Waliser (2015)) between 1980 and 2017 (n=1814), and found that the average height of the 0°C isotherm during these days was about 3500 m (Fig. S2d). This puts the height of the 0°C isotherm for the thunderstorm event below the average.

**Technical Corrections**

Line 151 – I think "formed" should be "forming" or "allowing the formation of"

The sentence now reads, *"In the March 2019 storm, updrafts in the deep convective clouds, identified by overshooting cloud tops (Fig. S6), could have transported smaller droplets to above the freezing level, (below 700 hPa), potentially allowing for the formation of hail with a positive charge (Fig. 3c, 1b)."*

Line 152 – rephrase to something like "At the time closest to the peak of the event in Santa Barbara, dry air was entrained between 300 hPa and 200 hPa with winds reaching approximately 100 knots (Fig 3a)"

The sentence now reads, *"At the time closest to the peak of the event in Santa Barbara, dry air was entrained between 600 hPa and 400 hPa as well as in the upper-levels between 300 hPa and 200 hPa (Fig 3a), which could have enhanced downdrafts contributing in the formation of electrified hailstones."*

Line 160 – should be "warmer than its environment"

The sentence now reads, *"When the updrafted droplets and downdrafted hailstones collide, they can release latent heat, and potentially form graupel, a softer form of hail that is warmer than its environment."*

Line 174 – consider rephrasing to "The last peak of lightning frequency" or something like that

The sentence now reads, *"The last peak of lightning frequency occurred between 6 March 00 UTC and 06 UTC with approximately 3000 IC pulses and less than 1000 CG pulses centered at 34°N and 120°W (Fig. S7a)."*

**Author response to Anonymous Referee 3 for "Brief Communication: An Electrifying Atmospheric River: Understanding the Thunderstorm Event in Santa Barbara County during March 2019" by Deanna Nash and Leila M.V. Carvalho.**

Responses to reviewer comments are given in blue text. New or changed text is given in italics (bold italics for emphasis where noted)
* * *
**General Comments**

The manuscript analyzes an AR event that occurred on March 2019. Despite having a relatively low amount of precipitation (77.6mm in 30h) it was extraordinary amount of lightning strikes in the region that stands out. The topic of this study is of interest to be published and the manuscripts is in general well written however I have a few structural issues that need to deal with before the manuscript is ready for publication.

We thank the reviewer for the time they took to review this paper and the constructive feedback that helped improve the paper, specifically with regard to the structure of the manuscript. Please see responses to comments below.

**Specific Comments**

I believe the entire Introduction section needs to be re-written. In the present form it's a mixture between introduction and results. Therefore, should have the following in mind when re-written the introduction: - What is an ARs; - Possible impacts and benefits (first paragraph should be in here); - Lightening brief introduction and precipitation measures and radar. - I would remove everything that is results from this specific event.

We have separated the introduction and results, and updated the introduction section to include a paragraph on the background of ARs and their relevance and impact to Southern California. We have also added a short background on lightning to the introduction.

Section 2. Parts of the introduction are already stated here. So, I would keep all the dataset and methodologies in this section and avoid repetition with the introduction.

Repetition with the introduction has been removed from the data and methods section and care has been taken to make sure this section is only dataset and methodologies and no results.

Section 3. I would include a new sub-section before sub-section "Extratropical Cyclone and AR Conditions". This new sub-section would be the description of the March 2019 event, with most of the information being taken what was already mentioned in the Introduction.

After restructuring the introduction (see above), we have moved the description of the March 2019 event to a new subsection in the results titled "March 2019 Event" to make sure the results are all in the results section.

**Minor Comments**

Figure 1. The color scales are a bit confusing.

A similar comment from RC2 was made. The colormaps in Fig. 1 were updated to only go to the maximum, and black and brown were removed from the colormaps.

How much do you trust in the vertical speed from reanalysis data?

We recognize that vertical velocity from reanalysis data is a calculated value. We found that the details provided from vertical velocity in the manuscript were not necessary to show updrafts and deep convection, so we removed vertical velocity from the manuscript and supplement. We have instead decided to focus on what the observed GOES-R infrared brightness temperature tells us, which is that there was an overshooting cloud top at approximately 4:30 UTC, indicating deep convection. We have updated Fig. S5 to highlight this information.

**List of changes made for "Brief Communication: An Electrifying Atmospheric River: Understanding the Thunderstorm Event in Santa Barbara County during March 2019" by Deanna Nash and Leila M.V. Carvalho.**

**Figure updates**

– Updated color bars on Fig. 1b, c, and d and changed precipitation data to NOAA NEXRAD L3 precipitation estimate for Fig. 1b.

– Added subplot c to Fig. 3 to show the equivalent potential temperature profile of the grid cell closest to Santa Barbara at the time closest to the peak of the storm.

– Fig. S2 removed since we decided to remove information on calculated vertical velocity and precipitation information was covered in Fig. 1b.

– Fig. S3 is now Fig. S2 and 3 additional subplots have been added showing the climatology of AR characteristics (IVT magnitude and direction and height of $0°$C isotherm) from past ARs in Santa Barbara.

– Fig. S5 is now Fig. S3 and has a zoomed out domain as well as IVT contour lines to indicate the spatial extent and intensity of the AR.

– Fig. S4 has been updated to show skew(t) - log(p) and equivalent potential temperature profiles every 6 hours for the grid cell that has the highest lightning flash density. CAPE is shown in the plot that indicates the location of the highest lightning flash density.

– Fig. S6 has been renamed to Fig. S5 and a zoomed inset map has been added as well as locations of NOAA NEXRAD L3 hail.

– Fig. S7 is now Fig. S6. The colorbar for Fig. S6c, S6d, and S6e has been updated to match Fig. 1c and 1d.

– Fig. S8 is now Fig. S7

– Fig. S9 is now Fig. S8

**Manuscript revisions**

– **Abstract**: The abstract has been updated to show the most important results of this study.

– **Introduction**: The introduction has been restructured so that the description of the March 2019 event was moved to a new results section titled, "March 2019 Event". In addition, a brief description of ARs and lightning has been added to the introduction.

– **Data and Methods**: Revisions have been made to the Data and Methods section to only discuss data sources and methods used for the study.

– **Results,** *March 2019 Event*: This new section in the results was added to describe the precipitation and lightning that occurred during the March 2019 event.

– **Results,** *Extratropical Cyclone and AR Conditions*: This section was updated to only discuss the synoptic conditions associated with the March 2019 event.

– **Results,** *Thermodynamic Conditions*: This section has been renamed (previously "Precipitation and Hail) since the main focus of this section is to describe the thermodynamic characteristics of the March 2019 event. This section has been revised to clarify the unique thermodynamic conditions of this event, focusing on equivalent potential temperature and the significantly above average water vapor flux content in the middle troposphere provided by the AR.

– **Results,** *Lightning Conditions*: This section has been edited to clarify the difference between what previous literature suggests are important factors for electrification and what the data implies about electrification for this particular storm.

– **Conclusions**: This section has been updated to summarize the findings of this study, which are that we found that it is possible that these thousands of lightning flashes that occurred in under a few hours were related to an AR that was characterized by an unusual deep moist layer extending from low-to-mid troposphere in an environment with potential instability and low elevation freezing level.

275

[revised manuscript text omitted]
. (b) Mean IVT of the AR objects that made landfall in Santa Barbara in all the months (blue lines) and only March (grey lines) between January 1980 and May 2019 based on the AR Catalog from Guan and Waliser (2015). The mean IVT for the AR Event on March 5 is shown by the red solid line. The means of the distributions are shown in the dotted line. (c) Same as (b) but for direction of mean IVT propagation (azimuth is 0° if IVT is directed to the north). (d) Same as (b) but for the height of the 0°C Isotherm (m) interpolated from MERRA2 temperature and geopotential height

[Figure]

**Figure S3.** CFSv2  850 hPa Equivalent Potential Temperature (shaded; K),  850 hPa winds (barbs; knots), and IVT greater than 250 kg m⁻¹ s⁻¹ ( white contours; every  250 kg m⁻¹ s⁻¹)  for each 6-hour time step between 4 March 2019 18 UTC and 6 March 2019 18 UTC.

[Figure]

**Figure S4.** *(left panel)* Skew(t) - log(p) vertical profile of CFSv2  temperature ( red line) and  dew point (green line) at the grid cell with the highest flash density (per 6 hours); *(right top panel)* CFSv2 CAPE (shaded, J kg⁻¹) and  MSLP (black dashed contours; hPa) with the location of the highest flash density indicated by the red dot; *(right bottom panel)* CFSv2 Equivalent Potential Temperature (blue line; K) at the grid cell with the highest flash density for each 6-hour time step between (a) 4 March 2019 18 UTC and (i) 6 March 2019 18 UTC.

[Figure]

**Figure S5.** Infrared brightness temperatures (shaded, °C) derived from band 13 of the GOES17 ABI L2 Cloud and Moisture Imagery Brightness Temperature at 6 March 2019 4:24 UTC. Detailed infrared brightness temperatures around Santa Barbara (outlined in red) are shown in the top left area of the map. Locations of NOAA NEXRAD L3 Hail Signatures (black points) identified between 4:15 UTC and 4:45 UTC on 6 March 2019 are shown on the inset map.

[Figure]

**Figure S6.** (a) ENGLN number of flashes per 15 minutes between 4 March 2019 0 UTC and 7 March 0 UTC for in-cloud (IC) flashes (blue line) and cloud-to-ground (CG) flashes (red line). (b) ENGLN average IC lightning flash height (green line; m) between 4 March 2019 0 UTC and 7 March 0 UTC. (c) ENGLN lightning flash count (shaded, flashes day$^{-1}$) interpolated to 0.1°and IVT greater than 250 kg m$^{-1}$ s$^{-1}$ (grey contours; every 100 kg m$^{-1}$ s$^{-1}$) for the 24-hour period of 4 March 2019. (d) Same as (c), but for the 24-hour period of 5 March 2019. (e) Same as (c), but for the 24-hour period of 6 March 2019.

[Figure]

**Figure S7.** GOES ABI L2 ACHC cloud top height (shaded; m) and the location of the majority of lightning flash points (red polygon) at each 6-hour time step between 4 March 2019 18 UTC and 6 March 2019 18 UTC.

[Figure]

**Figure S8.** CFSv2 height of $0°$ isotherm (shaded; m) and the location of the majority of lightning flash points (red polygon) at each 6-hour time step between 4 March 2019 18 UTC and 6 March 2019 18 UTC.

---

## Author Response (AR2)

**Minor Revision Author response to Anonymous Referee 2 for "Brief Communication: An Electrifying Atmospheric River: Understanding the Thunderstorm Event in Santa Barbara County during March 2019" by Deanna Nash and Leila M.V. Carvalho**

Responses to reviewer comments are given in blue text. New or changed text is given in italics (bold italics for emphasis where noted)
* * *
**General Comments**

General Comments

I am generally satisfied with the revisions produced by the authors. One minor comment is below along with some examples of proofing errors that should be checked.

We thank the reviewer for the time spent to review this manuscript and suggestions for minor revisions which improved the paper. Please see responses to specific comments below.

**Specific Comments**

Abstract and elsewhere – I don't think the lat/lon box noted should be called "Southern California". It is much larger. It seems like there would be similar results if you looked at lightning anomalies just below 36N for example (even if you will want to look at other quantities over a larger region to determine larger scale circulations that led to the local lightning extremes).

We agree with the reviewer that this region should not be called "southern California". We reviewed the statistics from the lightning data and found that when we looked at lightning from 30N to 37N and 130W to 115W the numbers were slightly different but no less significant. We have updated the manuscript to include the results from the new subregion that we hope the reviewer agrees is considered around Southern California. For example, the abstract now reads, "*The Earth Networks Global Lightning Network (ENGLN) detected 8,811 lightning flashes around southern California ($30°N$ to $37°N$ and $130°W$ to $115°W$) in 24 hours, which is roughly 2,500 times the climatological flash rate in this region.*"

Fig 3a text on low left is hard to read, consider adding partially transparent white box as in other overlaid legends on this figure.

We agree with the reviewer and have updated the figure so that the text in the bottom left corner of Fig 3a is easier to read.

Line 151 reference should be to figure S2d.

We have updated this line to reference the correct figure.

Some (minor) additional proofreading is needed, a couple examples below:

Line 33 – should be "a critical role"

Line 195 – verb should be singular to agree with "accumulated rainfall"

We have updated these two sentences and have proofread the manuscript for other minor grammatical issues.

**Minor Revision List of changes made for "Brief Communication: An Electrifying Atmospheric River: Understanding the Thunderstorm Event in Santa Barbara County during March 2019" by Deanna Nash and Leila M.V. Carvalho**

**Figure updates**

35      – Updated the text in the bottom left corner of Fig. 3a to increase readability.

**Manuscript revisions**

     – We have updated sections that refer to the number of lightning strikes (abstract, results, conclusions) within the region of Southern California, now considered (30°N to 37°N and 130°W to 115°W).

     – We have made sure to correct any incorrectly referenced figures.

40      – Other minor grammatical corrections have been made throughout the paper.

[revised manuscript text omitted]